# Factors Influencing Bank Customers' Orientations toward Islamic Banks: Indonesian Banking Perspective

**Krisna Nugraha \***, **Muhtosim Arief, Sri Bramantoro Abdinagoro and Pantri Heriyati**

Research in Management, Doctorate Program, BINUS Business School, Bina Nusantara University, Jakarta 11480, Indonesia
\* Correspondence: knugraha@gmail.com; Tel.: +62-818-247247

**Abstract:** During the COVID-19 pandemic, the Indonesian banking industry showed positive performance, high profitability, sustainable growth, and stability. Islamic banks grew by 9.50% and had a market share of 6.52 percent as of September 2021. This study aims to examine the industrial sector perspective on the factors that prevent consumers from becoming customers of Islamic banks, in particular the factors that influence consumer decisions not to become Sharia bank customers. This study used descriptive qualitative methods and in-depth interviews to confirm and obtain input from industry representatives regarding Islamic banks. Primary data collection was performed using a purposive sampling technique. Informants were head office officials, managers, heads of individual retail product development units, product features and policies, market education, marketing, customer acquisition, and individual retailers. The findings of this study are the existence of passive resistance of consumers to become customers of Islamic banks. In addition, there are obstacles for prospective customers of Islamic banks in responding to marketing stimuli due to the perception of risk, image, and weak marketing reach that have dominated passive resistance to Islamic banks (blocking effects). As a result, consumers prefer conventional banks and are less interested in becoming customers of Islamic banks. This means that there is no opposition to Islamic banks.

**Keywords:** holding; blocking; consumer resistance; service adoption; Islamic banks; banking perspective



## 1. Introduction

Despite being hit by the COVID-19 pandemic, the banking industry in Indonesia has shown positive performance, high profitability, sustainable growth, and increased stability. This growth includes both conventional banks and Islamic banks. State-owned enterprise bank (BUMN) loans grew by 0.63% and regional development banks (BPD) grew by 5.22%, and Sharia banks grew by 9.50%. Sharia banking market share as of September 2020 is 6.24 percent [1] while the market share of Islamic banking as of September 2021 is 6.52 percent [2]. This shows a positive growth of Islamic banking market share. Based on data from the Financial Services Authority (hereinafter referred to as OJK) that Assets, Disbursed Financing (PYD), and Third Party Funds (DPK) of Islamic banking until September 2021 showed positive developments (OJK, 2021). Based on the Deposit Insurance Agency report that Third Party Funds (DPK) in May 2021 amounted to Rp 6819.9 trillion. There was an increase of 0.6% compared to the previous month and an increase of 10.8% compared to May 2020.

Using traditional bank services as stated in the Al-Quran, Muslims should not be engaging with riba (usury), which is regarded a transgression. Indonesia National Ulama Council (MUI) has also released a fatwa No.1/2004 ruling about usury practice in conventional financial practices, which explicitly defines the practice of interest in money is a form of usury, and usury is haram (illegal from Islamic law). The practice of interest is haram, whether it is carried out by banks, insurance, capital markets, pawnshops, cooperatives, and other financial institutions or by individuals. The fatwa dictated that for Muslim

consumers, transactions based on interest calculations are not allowed in the areas that already have offices/networks of Sharia Financial Institutions and are easily accessible. From business perspectives, the main principles of Islamic finance are that: Wealth must be generated from legitimate trade and asset-based investment (the use of money for the purposes of making money is expressly forbidden). Investment should also have a social and an ethical benefit to wider society beyond pure return. With these foundations, benefits, and the large Muslim population in the country, the government of the Republic of Indonesia is taking strategic and tactical efforts to promote Islamic financial practices as an alternative to conventional systems, this includes opening the first syariah bank in 1993, refining financial regulation to align with syariah law, and the latest in 2020 establishing KNKS (National Committee of Syariah Economy and Financial Systems).

Still, with the above foundations, the growth of Islamic banks is still low or not comparable to the growth of conventional banks. The number of conventional bank customers continues to grow, exceeding the number of Islamic bank customers [3]. Similar to Indonesia, which has the largest Muslim population in the world, Islamic banks are still having difficulty increasing their market share even in countries with a majority Muslim population such as Egypt [4,5], Oman [6], Turkiye [7], Malaysia [8], Pakistan [9,10], and Indonesia [11–13]. Muslim consumers in nations with a majority Muslim population are not yet interested in becoming customers of Islamic banks, even though Islamic banks already have distinct advantages over regular banks and that Islamic law mandates that they do so [14–21].

The contrast with conventional banks can be seen from the comparison of the proportion of public deposits in large conventional banks with a ratio of more than 93% (Rp 6.421 trillion). While the ratio of Islamic banks is only 6 percent. Likewise, the number of conventional bank accounts reached 89 percent, while the number of Islamic bank accounts is only about 10 percent or as many as 37 million accounts. This fact shows that the Indonesian Muslim community has a higher level of trust in conventional banks compared to Islamic banks [22–25]. Even though most know that Islamic banks are run using Islamic financial principles, especially not using usury (*riba*) [26] which is banned in Islam [27,28]. A major principle of the Islamic economic system is risk sharing [29]. Islamic bank financial instruments have received less attention from the public [30–33].

Cases of weak Islamic banking also occur in the Turkish banking sector because Islamic bank loans do not have a significant effect on improving the economy and industry in Turkey [34]. Malaysians use Islamic banking products and services less frequently than conventional banking products and services [35]. The lack of interest in Islamic banks in the two countries is almost the same as in Indonesia, but the problems are different.

This phenomenon occurs because public awareness of Islamic banks is still low [36–38]. On the other hand, several studies have shown that Islamic banks are quite attractive in various countries with non-dominant Muslim populations, for instance, the interest of non-Muslim consumers in South Korea towards the Islamic finance industry, especially takaful products [39], consumer interest in becoming a customer of Islamic banks in the United States [40], and consumer interest in becoming a customer of Islamic banks in New Zealand [41]. This means that the tendency to believe does not always influence the decision to choose a bank. According to research, one of the most important factors is the availability of digital-based services [42,43]. Actually, Islamic banks have the same opportunities as conventional banks to enter the Indonesian banking market.

According to [44–46] that the reason a consumer refuses a marketing offer is not the same as the reason a consumer accepts a marketing offer. In the context of Islamic banking, the understanding of consumer resistance who is not a user or bank customer has a different perspective to market players. The findings of this study are expected to explain the factors that cause the low number of Islamic bank customers in Indonesia.

According to the researcher, the theory of consumer resistance can be used in this study because it has never been used to study Islamic banks, and at the same time to complement the limitations of previous research studies [47–49]. Seth et al. [50] conducted

research on the intersection between consumer resistance and consumer inertia, which form resistance to change and strengthen consumers' desire to maintain the status quo.

The application of consumer resistance theory can describe various factors that cause consumers to refuse to adopt or switch to other banking products or service providers. However, the use of Innovation Resistance theory in previous studies is independent of consumer switching theory [51,52]. This concept of consumer switching excels in predicting the intentions and behaviors that drive consumer switching [52] but has not yet discussed the role of the bond (satisfaction and loyalty) between the consumer and the product or service provider as a holding factor. movement or shift of consumers [53,54].

This concept also does not take into account the disruption of innovation that diverts consumers from their original goals, as has happened in the banking industry with the presence of financial technology (Fintech) companies [55,56]. FinTech is reshaping the financial services industry [57]. The use of computer-based technologies has revolutionized the financial services industry [58], utilizing technology to deliver new and improved financial services [59]. Typically, mobile payment products are referred to [60]. Fintech is a financial sector paradigm shift and the embodiment of technological advances and digital transformation [61]. Christensen defines disruption as a theory of competitive response. Disruption is a process, not an event, and innovations can be disruptive only in comparison to something else [62]. The characteristics of disruptive innovation are simpler, more convenient, and cheaper products.

Based on this explanation, the researcher adopts two concepts of consumer resistance and consumer switching [63–65] in an attempt to fill research gaps that cause barriers to product or service adoption. In addition, it examines the quality of the relationship between consumers and current product or service providers (holding factors) as well as negative attitudes towards alternative products or services that can affect barriers to adoption of a product or service (blocking factors) [66,67].

The purpose of this study is to determine the perspective of entrepreneurs regarding the factors that prevent consumers from becoming customers of Islamic banks. In addition, this study intends to examine the industry's perspective on the factors that influence consumer decisions not to become customers of Islamic banks. This research also seeks input from entrepreneurs regarding appropriate marketing strategies, product and process innovations, and obstacles in implementing these marketing and innovation strategies.

## 2. Literature Review

### 2.1. Holding Effects

This variable is conceptually defined as prospective Islamic bank customers' attachment to conventional banks used today due to long-standing relationships and satisfaction with service. Furthermore, they have experience with receiving good service quality, high trust in the bank, and the consumer's aversion to conventional banks, which causes reluctance to become customers of Islamic banks. The first dimension is customer satisfaction with conventional banks (satisfaction with the current bank), which is measured using three indicators: satisfaction with bank services in general, satisfaction with automatic teller machine (ATM) services, namely ATM facilities that are many and can always be used. In addition, feeling satisfaction with the service of bank officers, including in the welcome upon arrival, making transactions with tellers, or consulting with customer service.

The second dimension is the service quality of conventional banks or banks that are being used, which is measured using three indicators, namely service quality related to the ease with which customers interact with banks because bank branch offices are easily accessible, service quality related to the ease of interacting with banks because ATMs are easily accessible, and service quality related to the ease of interacting with bank officers.

The third dimension is trust in conventional banks or banks that are used today, which is measured using four indicators: the integrity of bank employees, the competence of bank employees, security of customer money, namely the bank's ability to store cus-

tomer's money in accordance with the amount, and the bank's ability to provide solutions to problems.

This variable's fourth dimension is customer aversion to traditional banks, interaction habits, and interest in alternative services [68–72].

While the conceptual definition of the variable Adoption Resistance is the attitude of prospective customers of Islamic banks toward individual savings products of Islamic banks due to switching costs (switching from conventional banks to Islamic banks), the tendency to maintain current banking habits, and the tendency to choose banks rationally based on the number of branch offices and ATMs. This variable has no dimensions and is measured using five indicators: switching hassle, switching benefits, habit maintenance, selecting a bank with a large number of branch offices, and selecting a bank with a large number of ATMs [73–77].

According to [53,54] studies on the adoption of mobile payment solutions were dominated by interest or intention to adopt, with only four studies on resistance. According to [50,77], studies on Online Travel Agents (OTA) generally question the interest or intention to adopt, and resistance is rarely discussed. Meanwhile, according to [78] research on cloud computing adoption, the information technology literature generally prioritizes factors that influence interest in using and accepting technology, but research on resistance to technology application is very limited. According to [79] research on e-commerce, research on consumer behavior towards e-commerce is rarely aimed at understanding consumer resistance behavior that ultimately decides not to use e-commerce services. This study, in accordance with the researcher's understanding from previous studies, constructs a decision construct not to adopt a product or service as the dependent variable.

### 2.2. Blocking Effects

Variable blocking effects are defined as an impediment to prospective Islamic bank customers responding to Islamic bank marketing stimulation due to risk perception, image, and the limited reach of Islamic bank marketing. Islamic banks have attempted a variety of marketing activities, but the responses from potential customers have not always been positive. This variable has three dimensions: the risk dimension, which is measured by four indicators, namely financial gain, the size of the Sharia bank's business, the number of ATMs in Islamic banks, which is small compared to conventional banks, and the number of friends and family who become bank customers. Concerns have been raised about transaction difficulties between friends and family, for example, having to pay interbank transfer fees [8,9,80–83].

### 2.3. Consumer Resistance

Innovation resistance as a concept is relatively new [84,85]. There has been very little consideration dedicated to the causes behind customer resistance to innovation [86]. Consumers are concerned about physical, social, and economic risks. They are also concerned about the unclear performance of alternative products/services, as well as potential side effects or other unknown consequences [73–76,87]. The study on resistance to innovation by [74] introduces the concepts of passive resistance and active resistance, which are the reasons why consumers do not adopt the innovations offered by marketers. Passively resistant consumers are not open to any sort of product innovation that they believe would impose change or imperil their status quo and will not engage in any product assessments, but rather reject the innovation at the knowledge status [86].

According to the research of [50], a strong habit of doing what is currently carried out greatly inhibits consumers' desire to change, including seeking alternative information products/services, considering and using new products/services. Consumers are also deterred from changing and adopting innovations because they are concerned about the risks associated with change or innovation. Factors that discourage consumers from using products or services are influenced not only by the resistance factor but also by the binding According to Zavestoski (2002) anti-consumption is defined as "a resistance to, disgust for,

or even resentment towards consumption [88]. Adoption and resistance are not always synonymous [89]. Resistance to change may be described by several human qualities that can be applicable not just to employees but also, most likely, to individuals and, therefore, to customers [90].

## 3. Methodology

Considering that research on rejection of Islamic banks, namely the holding effect, blocking effect, and consumer resistance is a new topic and the motive for the rejection is consumer subjectivity, the authors used descriptive qualitative research that is exploratory and inductive. Sometimes the terms qualitative research and descriptive research are used interchangeably [91] using an inductive approach [92,93]. Denzin and Lincoln describe qualitative research as an interpretive, naturalistic approach to the world [94].

This research used qualitative and descriptive research methods which are widely used to research various disciplines such as education, psychology, and social sciences [91] which focuses on a detailed description of the context and frequently emerges from a situation problem [95]. The process of qualitative research is inductive, which means that the researcher collects data to build a concept, hypothesis, or theory [96]. Three related concepts of qualitative methods: (1) non-positivistic qualitative research epistemologies; (2) qualitative research strategies that focus on interpreting or revealing meanings rather than generalizing causal relationships; and (3) qualitative research techniques that do not rely on numbers (e.g., interviewing) [97].

The in-depth interviews used in this study aim to confirm and obtain input from industry representatives, which include informants from conventional banks, Islamic banks, and mobile-based electronic money application service providers. The question is, namely the view of industry players on the factors that prevent consumers from becoming customers of Islamic banks. In addition, the interview aims to determine the industry's perspective on the factors that influence consumer decisions not to become customers of Islamic banks. In addition, this research also seeks input from industry representative regarding marketing strategies that can be applied, product innovations that can be developed, and obstacles in implementing these strategies and/or innovations. The following are interview guidelines utilized for data collection:

**In-Depth Interview Guide to Conventional Bank Participants (Holding Effects)**

1. Elaborate the main factors that keep customers continuing relationship with the bank.
2. What is the view of Islamic banks in general?
3. What would be the reasons for not being a customer of a sharia bank?
4. Discuss how conventional bank keeps customer satisfy.
5. Discuss how conventional bank ensure services quality.
6. Discuss how conventional bank maintain customer trusts.
7. Discuss the type of of customer inertia towards conventional banks.

**In-Depth Interview Guide to Sharia Bank Participants (Blokcing Effects)**

1. What is the view of Islamic banks in general?
2. What would be the reasons for not being a customer of a sharia bank?
3. Discuss the potential risk of becoming a customer of a sharia bank.
4. Discuss the negative image of Islamic banks.
5. Discuss the marketing weakness of Islamic banks.

**In-Depth Interview Guide to Mobile Based Electronic Money Application Services Providers (Diverting Effects)**

1. What is the view of Islamic banks in general?
2. What would be the reasons for not being a customer of a sharia bank?
3. Discuss your innovation to complete banking services.
4. Discuss your marketing attractiveness.
5. Discuss your ecosystems strategies.

6.   Discuss your services economic values.

Participants in this interview (Table 1) are head office officials with a minimum of 5 years of work experience in the areas of individual retail product development tasks, product features and policies, market education, marketing, and individual retail customer acquisition, as listed below:

**Table 1.** In-depth interview participant.

| Industry | Company | Title | Code |
|---|---|---|---|
| State Owned Enterprise Syariah Bank | PT. Bank Syariah Indonesia, Tbk | Group Head Funding Retail Hajj and Umrah | BSI1 |
| | | Department Head Plan and Evaluation Retail Funding | BSI2 |
| | | Team Leader Product Development Retail Funding | BSI3 |
| National Syariah Bank | PT. Bank BCA Syariah | Division Head Business and Communication | BCS1 |
| | | Department Head Corporate Communication | BCS2 |
| | PT. Bank Mega Syariah | Department Head Retail Product Development | MES1 |
| | | Marketing Manager Strategy and Business Policy | MES2 |
| | PT. Bank Muamalat | Head of Product Specialist, Retail liabilities, Wealth Management and Digital Banking | BMI1 |
| | | Senior Relationship Manager Retail Funding | BMI2 |
| State Owned Conventional Bank | PT. Bank Rakyat Indonesia, Tbk | Department Head Retail Funding Customer Acquisition | BRI1 |
| | | Department Head Retail Funding Product Development | BRI2 |
| National Conventional Bank | PT. Bank Central Asia, Tbk | Head of Retail Funding Customer Acquisition | BCA1 |
| | | Branch Operations Head and Retail Funding | BCA2 |
| E-Money Application Services Provider | PT. Dompet Anak Bangsa (GoPay) | Payment Product Development Manager | GOP1 |
| | | Customer Acquisition Manager | GOP2 |
| | PT. Visionet Internasional (OVO) | Marketing manager Retail and Merchant Expansion | OVO1 |
| | | Operational Manager Retail Outlet Support | OVO2 |

All participants received and understood all the information they needed to decide whether they want to participate. This included information about the background of the study, the study's benefits, funding, and institutional approval. Informants were selected by the research agency based on the criteria provided. There was no relationship between researcher and participants.

Primary data collection using a purposive or judgment sampling technique. Purposive sampling, also known as assessment sampling, is the deliberate selection of participants based on the qualities that participants possess [98] which is intended to focus on people with certain characteristics who would be better able to assist relevant research [98] use to select respondents who are most likely to provide relevant and useful information [99]. Because the respondents were asked the same questions in the same order, the results are expected to be consistent. Meanwhile, comparative analysis was the primary function used to analyze the data in order to comprehend the relationship between all tested variables.

This technique was carried out with the help of an interview guide, which is divided into three sections: (1) research topics and objectives, (2) research variables relevant to the participant industry, and (3) variable dimensions and indicators relevant to the participant industry. The detailed interview guide sheet is attached to this manuscript as an appendix. All interview activities were audio and video recorded for later use as the foundation

for interview transcripts and interview summaries. An independent party listened to the recorded interviews and prepared (1) verbatim transcript documents per interview session and (2) narrative documents from the interviews. By conveying direct quotes from interviews, the findings of quantitative data processing are strengthened.

The collected data were then reviewed, classified, clarified, synthesized, and elaborated in an integrative manner using interpretive principles and logical explanations from stakeholders' experiences related to their knowledge and insight to comprehend the phenomenon of factors that prevent consumers from becoming customers of Islamic banks [93]. This information was then evaluated using two methods: external criticism and internal criticism evaluation. External criticism was carried out by paying attention to the authenticity of the data obtained (data validation), whereas internal criticism was carried out to ensure the accuracy of the data (data reliability). These two evaluation approaches were used to discover facts and answer unanswered questions, particularly those concerning the factors that prevent consumers from becoming customers of Islamic banks.

## 4. Factors Influencing Bank Customers' Orientations toward Islamic Banks

This study divided the research findings into five sections during the discussion session. Each group is the result of an analysis of respondents' perspectives on the influence of various factors that prevent consumers from becoming customers of Islamic banks. Holding Effects, Blocking Effects, Adoption Resistance, Non-Adopting Decision, and Adoption Resistance are among the variables used in the interview instruments. Based on previous literature reviews, this variable was used to limit the scope of the research. Furthermore, the researchers asked respondents about the relationships between these variables in order to obtain a comprehensive and systematic picture of the factors that prevent consumers from becoming customers of Islamic banks.

### 4.1. The Influence of Holding Effects on Non-Adopting Decisions

According to Hati et al. [11], consumers who rationally evaluate the quality of their relationship with conventional banks in use today are discouraged from sacrificing convenience by switching to Islamic banks, regardless of the Islamic principles used.

Regarding trust in traditional banks, BRI1 participants stated: " . . . the belief that serves as the foundation . . . It takes time to process something new . . . "

This statement reflects respondents' belief that traditional banks used today are run by competent professionals, which fosters customer trust. According to Wasan [100]'s research, the competence of bank employees is an important factor in creating a positive customer experience. This is also true for customers in Indonesia, where positive interactions with traditional bank employees are attributed to good bank employee competencies. Similarly, Kamarudin & Kassim [101]'s explanation that emphasizes the professionalism of bank employees has an effect on customer satisfaction.

With their financial capabilities, traditional banks are able to maintain and improve the competence and professionalism of their employees, particularly those who deal directly with customers. This undoubtedly makes a favorable impression on customers and strengthens their attachment to the bank. Because they lack trust in Islamic banks as a result of their trust in conventional banks, people are less likely to become customers of Islamic banks.

Another study [102] found that the length of time that causes consumer attachment to current service providers influences consumers' low desire to view or seek alternative services. Even when there are alternatives, consumers are often uninterested in learning more, or they compare what is currently accepted and conclude that the service they are currently using is superior to the alternative service.

### 4.2. The Influence of Holding Effects on Consumer Resistance

The interest in switching customers to Islamic banks is related to the high quality of service provided by conventional banks today. According to the findings of a study

conducted in Kuwait by [103] on consumer switching behavior in Islamic banks, service reliability and service convenience are positively related and significantly increase customer satisfaction, which influences customers' willingness to switch to other banks. The same thing happened to most conventional bank customers in Indonesia, who had been using conventional bank services without incident for a long time.

Meanwhile, according to Shankar & Kumari [104]'s research on the adoption of electric vehicles from the seller's perspective, vehicle sellers' inertia forms resistance to not selling electric vehicles. The qualitative research of [105] on community resistance to mobile banking in Colombia demonstrates that the habit of transacting in the traditional manner forms consumer resistance to the use of mobile banking technology.

Based on Sultan [106]'s research on switching costs in the banking industry demonstrates the importance of the quality of customer relationships with banks in determining the level of switching costs. The higher the switching costs, the better the customer relationship with the bank. As a result, customers are hesitant to evaluate other alternative banking services, resulting in consumer resistance. According to Sultan [106]'s research, the quality of customer–bank relationships is a factor that influences consumer interest in choosing or switching banks. According to [106], the initial information that consumers seek and collect before choosing a bank is bank reputation, user environment, quality of bank services, and location of branch offices and ATMs. Consumers also believe that if their current bank is better than their intended bank, they are less likely to switch to a new bank, even if there are other benefits.

In an in-depth interview, Bank Rakyat Indonesia (hereinafter BRI), 1 participant stated that conventional banks create these quality customer relationships by being able to serve and support all customer needs as the customer world evolves.

> " . . . The second factor is satisfaction, which explains why there is a satisfied effect," the participants said. This is due to the ease with which it can serve and support all of its customers' needs. So, in my opinion, the most important factor is to meet all of the needs of the customers. In other words, you must be able to adapt to the most recent trends . . . "

Customers satisfaction built through quality services, according to Bank Central Asia (hereinafter BCA), 1 participant, is the mainstay of traditional banks. According to the participants:

> " . . . From what I've seen, it's not like the previous BCA bank jargon, where consumers get tired due to long lines . . . This means that we have advanced to a high level of technology. Previously, those who formed queues did so because there were a large number of customers who wanted to meet with customer service (hereinafter CS) and tellers in a single working day. Customers no longer need to visit the branch now that there is a CRM; instead, they can go to the machine and present their ID card. There will be no more queues . . . "

This is supported by the perspectives of BCA2 participants, who explained that traditional banks are always striving to improve service quality through the use of cutting-edge technology. As explained in the following:

> " . . . With the current technology, we are attempting to improve the quality of the inline . . . This means that every bank should be technologically advanced . . . "

The same message was conveyed by BCA1 participants, who stated, " . . . The findings of this study are consistent with what happens to our customers on a regular basis . . . This means that service is the most important factor for customers. They are unconcerned about the return on investment or the high interest income. Customers, in my opinion, are most likely to consider the services of a commercial bank or a traditional bank . . . "

This description is consistent with Mursal et al. [107]'s research, which indicates that service quality and customer convenience are important factors in the formation of high

value equity. The presence of high-value equity is closely related to customer loyalty, which keeps them from switching to other service providers.

Customers' resistance stems primarily from their close relationship with conventional banks, so they do not want to change their current conventional banking habits, believe that all banking needs have been met with conventional banks, and believe that having savings accounts with both conventional and Islamic banks will be difficult. Consumers who have used conventional bank services and believe that all of their banking needs have been met by conventional banks do not see the added value provided by Islamic banks, and thus do not see the need for a Sharia bank savings account.

*4.3. The Effect of Blocking on Non-Adopting Decisions*

Based on Sun et al. [108]'s study on unethical consumer attitudes demonstrates the strong influence of social norms caused by those closest to them, especially if the influence is already collective, where many people oppose a particular attitude or decision. This is related to the data from this study, which show that because the respondent's environment is not as a customer of a Sharia bank, the respondent believes that they do not receive support from their surroundings to become a customer of a Sharia bank.

According to the findings of this study, the Islamic banking industry still has a negative image among the Indonesian Muslim community. According to Bank Syariah Indonesia (hereinafter BSI) 1 participants, as follows:

> " . . . we are quite late, and changing behavior takes time in order to be appealing to consumers . . . Sir, the last one is about perception. We don't know why, and it's possible that we improved it inadvertently and easily, particularly before 2016. People still believe that Islamic banking is obsolete, backward, and out of date, sir . . . "

> " . . . The obstacle, in my opinion, is reverting back to the initial problem; . . . sometimes people don't care if it's Sharia or not; it's the same thing . . . " said one MES1 participant.

This statement reinforces the fact that for the Indonesian Muslim community, Islamic banks are no better than conventional banks, as stated in research [109] that most consumers believe that Islamic banks and conventional banks are the same. According to the following BSI3 participants, consumer perceptions of Islamic banks are inextricably linked to the Islamic finance industry's low literacy:

> " . . . it is true that literacy is a problem with Islamic banks . . . Especially since our current network is still very limited, and Alhamdulillah, we are still being helped by the services of our Syariah Bank or main banks to help improve literacy, albeit slowly . . . "

According to the following MES1 participants, " . . . consumer understanding today assumes that Islamic and conventional banks are the same . . . as a result, the Muslim community is very large in terms of quantity . . . On the other hand, there is a misunderstanding of the Islamic economy . . . "

Hati research et al. [11] discovered a link between consumer familiarity with the brand and brand trust. This condition influences consumers' willingness to adopt a brand. Of course, the greater the familiarity, the greater the consumer trust and attachment to the brand. This has not happened to prospective Islamic bank customers in Indonesia, where the level of familiarity with Islamic banks is not comparable to that of conventional banks. As a result, it remains difficult to establish trust in Islamic banks, which leads to consumers declining to become customers of Islamic banks.

According to [110], the existence of asymmetry in the delivery of marketing information can result in unclear consequences and benefits of using alternative services compared to services that are currently used. Well-informed people have the raw materials for further processing and determining attitudes toward service innovation. Those who do not

receive enough information about service innovations are more likely to avoid trying or digging deeper.

### 4.4. The Influence of Blocking Effects on Consumer Resistance

The blocking effects and adoption resistance are inextricably linked. This means that the greater the blocking effects of the currently in use product or service, the greater the consumer's resistance to trying or using the product or service [111] discovered that the higher the negative perception, the less likely consumers are to use the products or services offered. Although there are three types of consumer decisions, they are rejection which tends to be negative by inviting others to do the same, opposition which refuses without inviting others to do the same, and delays. Furthermore, according to this study, a negative product or service image has a direct effect on rejection attitudes.

According to [112–114] research, consumers' stereotypical thinking about new innovations or service offerings leads to the formation of a negative image of the new innovation or service. A negative attitude, which is a form of consumer resistance to marketing offers, is most strongly influenced by a negative image. When compared to usage or functional barriers, consumer resistance is primarily formed due to a negative image.

Bank Muamalat Indonesia (hereinafter BMI), 1 participant expresses the following in this regard: " . . . Yes, we believe that marketing should be strengthened. For example, many of our features are unknown to our customers . . . "

Marketing communication is important, according to the same participants, and is recognized as a shortage of Islamic banks, as follows:

" . . . So the most important thing is communication and marketing from the bank . . . ".

Participants in BSI1 expressed the same sentiment, namely: " . . . Yes, we may have a problem; we have public relations in communication . . . So it's true, sir, that the main thing you're researching now is Islamic financial literacy, which is desperately needed by the public today so that they know what's wrong and what's right . . . "

Concerning the shortcomings of Islamic bank marketing communications, BCS2 participants stated, " . . . I completely agree with the issues of information and communication that must be improved, and this, of course, cannot be done alone . . . The industry must also promote Sharia-compliant content. Sharia, especially since we are in Indonesia, should be preferable to conventional . . . "

" . . . not communicating advertising, yes, it's not as intensive as Bank Mandiri and Bank BCA," said BMI1 participants. " . . . Let us be honest about this. That is how Islamic banks market their products . . . "

BMI2 participants from the same Islamic bank stated the following: " . . . What is the biggest problem with fellow Islamic banks? . . . It is, indeed, more of a promotion . . . and promotion costs money. When deciding where to put an ATM or where to open a branch, the cost of the rental must be factored in. So, how about the ROI? Do not allow us to go over budget, and then do not return the capital. Yes, that could be an issue for the employees because they don't make a profit . . . "

According to the descriptions provided by several participants representing Islamic banks, it is recognized that Islamic banks have marketing flaws, which cause people to have a broad understanding of what and how Islamic banks operate. One example of an Islamic bank's marketing weakness is that it does not widely communicate that Islamic bank customers can use conventional bank ATMs for free. This is despite the fact that the presence of a large number of ATMs is an important attraction for prospective Islamic bank customers. According to respondents, the first reason for not being a Sharia bank customer is the limited number of ATMs and branch offices of Islamic banks.

This does not have to happen because BCA2 participants stated that Islamic bank customers' ATM cards can be used at conventional bank ATMs, as follows: " . . . For my

product (BCA bank), there is BCA Syariah, but if you want to withdraw an account at an ATM, it has been merged with Conventional Banks . . . "

" . . . So in terms of features, Alhamdulillah, Bank Syariah Indonesia cash withdrawals at all our main ATMs at Bank Mandiri are free, sir," said one BSI1 participant. The information was not made public . . . "

Other communication flaws that were clearly communicated by BCS2 interview participants in relation to the joint use of conventional bank ATMs with Islamic banks in the same business group are as follows:

" . . . According to licensing in our banking sector in Indonesia, the emerging brand is the acquirer's brand . . . As a result, we expect to see the BCA Syariah logo there as well. This, however, cannot be accepted in an orderly fashion, but we hope that it will lead to that. When we were there, there was a licensing issue, whether BCA Syariah already had an acquirer license; I wouldn't go so far as to say it's a regulatory impediment because we have never directly stated to the regulator whether it could be like that . . . "

This explanation demonstrates that there are provisions that restrict the placement of Islamic bank logos on conventional bank ATM machines in one business group. As a result, despite the fact that the infrastructure is already in place and ready for use, Islamic banks are having difficulty attracting the attention of potential customers. Furthermore, participants in BCS2 described the following:

" . . . when conventional banks withdraw from Aceh, customers must be convinced that BCA Syariah ATMs can be used at BCA ATMs . . . Due to the lack of funds, the BCA Syariah logo appeared, and a digital BCA was added to the ATM machines at the same time. Bank BCA has only one implementation in Aceh in terms of *Qanun* implementation, but there is also a blessing. It's just that it can't be big because there is still an acquirer, who is important, but this is interesting because we know that when we see it, people believe . . . "

Participants in BSI1 agreed on the following statement: " . . . The central company also has a business . . . This is a pending task for us, and we appreciate your input, sir. This means that we believe that if all of the Parent Bank's ATMs read "Free Cash Withdrawals for Sharia Bank Customers," that's a 'wow,' sir. That is what we are still negotiating with the central company, sir . . . "

Islamic banking industry participants are optimistic about the provisions for displaying the Islamic bank logo on conventional bank ATM machines. According to the following BCS2 participants, this issue has even become a separate discussion within the Islamic banking industry:

" . . . After seeing it, we hope that in the future, if possible, there will be a BCA Syariah logo . . . A BCA subsidiary, for example, at a BCA ATM machine, will greatly encourage people to join BCA. This is also consistent with our conversations with Bank Syariah Mandiri (hereinafter BSM) colleagues. BSI is the same; coincidentally, a BSI friend, ex-BSM, stated that the provisions are similar . . . "

The provisions regarding the placement of this logo in the Islamic banking industry should be a top–down provision from the regulator to the industry. Specifically, to provide genuine support for Islamic banks, not just in terms of capital and risk management. However, there is also a marketing aspect that encourages customer acquisition. This was conveyed by BCS2 participants in the following way:

" . . . This is a direct topic that can be resolved if everyone includes genuine support for Islamic banking . . . Because, according to the facts on the ground, Islamic banks are not going away. As for the customers we're asking now, because they're early adopters, they believe the bank must have an ATM. Because, after all, money is still important. Second, it is a form of presence, a close-to-them

service. Even if it's just a machine, they believe it's important that there's an ATM and a branch . . . "

However, this does not rule out the possibility of Islamic banks developing good marketing communication strategies because, according to BCS2 participants, there are provisions for conveying marketing messages that are different from conventional banks in accordance with Islamic Sharia principles, so there is actually a gap to convey messages. recently stated the appeal of Islamic banks Participants in BCS2 express this as follows:

" . . . So the OJK wanted to invite us several times to socialize about advertisements that are allowed and not allowed . . . So, when it comes to regulations, there are some, but they are not impediments. There is one more rule we must follow, and that is the DSN Indonesian Uelama Council (hereinafter MUI) *fatwa*. There is a DSN Indonesian Uelama Council (hereinafter MUI) *fatwa* that I'd like to mention in the forum; it forces us to be more creative, not to hinder, but to be more creative . . . As in a fatwa granting a reward for Sharia. The *Wadiah* contract is a gift that can be given before the contract is signed. There is a fatwa like this. We use *Wadiah* in savings products, which means we can give gifts before people open savings, i.e., pay, vouchers. As a result, we are not like traditional banks, which may invite us to increase our balance and then give us something. So, sir, we can't do that. So it's just more difficult . . . there are still regulations that we have to follow, but we're not saying they're obstacles. But it forces us to be more astute because there are numerous contracts, including *Wadiah* contracts, *Mudarobah* . . . This makes it difficult for us to separate because *Wadiah* contains provisions like that that are only promoted for *Mudarobah* savings. As a result of the abundance of gifts, people will be perplexed. So, if I come to the conclusion that the regulation exists and forces us to be creative . . . "

According to the statement, Islamic banks are subject to special religious law regulations. This is unavoidable, but it does not mean that Islamic banks cannot develop creative communication while adhering to Shari'a principles. However, the existence of these constraints necessitates greater efforts in order to carry out creative and appealing marketing, ensuring that there is no information asymmetry between the natural uniqueness of Islamic banks and consumer understanding.

### 4.5. The Effect of Consumer Resistance on Non-Adopting Decisions

According to [115,116]'s consumer resistance hierarchy, which consists of (a) postponement, (b) rejection, and (c) opposition, the decisions of Indonesian Muslim consumers not to become customers of Islamic banks fall into the following categories:

a.   Delay, by stating that he is not yet interested in becoming a customer of a Sharia bank: Indonesian Muslim consumers tend to put off becoming customers of Islamic banks.
b.   Rejection or rejection by stating that if you want to open a new bank account, you should go with a traditional bank.

According to information submitted by MES2 in-depth interview participants, Islamic bank enthusiasts are generally consumers over the age of 35, whereas millennial consumers have no interest in:

" . . . The level of awareness remains difficult for markets with a population of 35 or less . . . "

According to this statement, the general reason that consumers have not yet become customers of Islamic banks is that consumers are not yet interested in Islamic banks because Islamic banks do not have many branches and ATMs, consumers have never (very rarely) received information and knowledge about Islamic bank services/products, consumers believe Islamic banks have not thoroughly implemented Islamic law, and Sharia bank branch offices.

## 5. Implication

This study is the first to explore the possibility of using consumer switching theory and consumer resistance simultaneously to understand the reasons why consumers do not adopt the products or services offered. It is different from the previous theory of consumer switching proposed by Bansal [52], which only explains the factors that cause consumer switching, or different from the application of consumer resistance theory in general which only confirms the factors causing consumer resistance as researched by Kleijnen [115]. This study offers a new construct, namely holding effects, blocking effects, and diverting effects, which can be alternative antecedents of research on switching and consumer resistance. The study by M. C. Claudy [46] found that "reasons for and against adoption are not just opposites of each other; they are also qualitatively different constructs that affect consumers' decisions about adoption in different ways." That is, the reasons for adopting and the reasons for not adopting are not opposites, but each has its own unique reasons. This makes researchers surer that this research has room for new ideas that can lead to new knowledge. This study gives Islamic banks the first clues they need to figure out why they do not have many customers. Information from Islamic banks' competitors, such as traditional banks and fintech companies, gives objective information. Islamic banks at least know that people tend to put off becoming a customer of a sharia bank or choose traditional banks instead. This shows that there is resistance to Islamic banks among Muslim consumers. Islamic banks need to do a better job of getting their message across so that people understand that being a customer of a sharia bank is more than just fulfilling religious obligations. This is because Islamic banks are already different from other banks in many ways. Islamic banks also need to grow their presence in the community, both naturally and artificially, so that people feel like they know Islamic banks and are more likely to trust them and become customers.

## 6. Conclusions

According to the findings of the data analysis, Indonesian Muslim consumers are rational buyers who prioritize direct benefits from interactions with the financial services industry. Furthermore, knowledge of the MUI *fatwa* on *haram* interest has no effect on the reasons why consumers do not become customers of Islamic banks. This study also discovered that conventional banks have developed strong customer attachments, making Muslim consumers hesitant to switch to Islamic banks. On the other hand, Islamic banks have marketing flaws that have prevented them from attracting Muslim customers to become Islamic bank customers. Another consideration is how mobile-based electronic money applications can disrupt consumers' attention to traditional banking services (conventional or Sharia) as a supplement, but not as a replacement. Bank services are still required by customers.

As a result, this study concludes that consumers have a passive resistance to becoming customers of Islamic banks. The existence of barriers to prospective Islamic bank customers responding to marketing stimulation of Islamic banks due to risk perception, image, and weakness of Islamic bank marketing reach dominates passive resistance to Islamic banks (blocking effects). As a result, consumers prefer traditional banks to open new savings accounts and are less interested in becoming customers of Islamic banks. Meaning that there is no opposition to Islamic banks.

## 7. Limitations and Future Studies

However, when compared to aggressive customer acquisition efforts, Islamic banks in general continue to strive and maintain financial ratios. Furthermore, regulators play an important role in assisting Islamic banks in improving their ability to attract prospective customers. The research hopes that there will be advancements in future research to gain more diverse knowledge that can enrich the treasures of knowledge in the fields of marketing and strategic management. According to Huang et al. [72]'s research, there are several areas that continue to receive less attention on the topic of consumer resistance.

This study was conducted during the process of merging state-owned Islamic banks into Indonesian Islamic Banks, so a complete picture of the impact of the presence of Indonesian Islamic Banks on consumers could not be obtained. Future research can be expanded to include new factors that emerge as a result of the existence of Bank Syariah Indonesia as Indonesia's largest Islamic bank.

**Author Contributions:** Supervision, M.A., S.B.A. and P.H.; Writing—original draft, K.N. All authors have read and agreed to the published version of the manuscript.

**Funding:** This research received no external funding.

**Institutional Review Board Statement:** Not applicable.

**Informed Consent Statement:** Informed consent was obtained from all subjects involved in the study.

**Data Availability Statement:** Not applicable.

**Conflicts of Interest:** The authors declare no conflict of interest.

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
