# Peer review of "Factors Influencing Bank Customers’ Orientations toward Islamic Banks: Indonesian Banking Perspective"

_sustainability, doi:10.3390/su141912506_

Round 1

Reviewer 1 Report

Research question is relevant and of great interest in the area of banking strategies. The adoption by some customers of a provider of financial products due to their membership to the same community is studied on several situation. in the indonesian context, at the introduction level, it's necessary to explain why it could be expected to see muslim people becoming customers of an islamic bank. what is the justification ? (several work exist based on the fact the customers shared with bank the same value-based behavior). that's whay it could suprising to see these customers don't move to islamic bank and justifiy the research to understand why?

Theorical frame work : the number and the relevance of references on this subject is the main positive point on this paper. the three concepts (holding effect, blocking effects consumer resistance) are relevant to explain the behavior of these customers and they produce switching cost enough significative to not incitate to prioritize the membership to a religious community. these three concepts are of course higly connected. i think in this part 2.  the author have to build a true conceptual framework considering the relationship between the concepts. the research couldn't be only on the fact that these three dimensions exist or not in the indonesian context.

Methodology : clearly the weakest part of the paper. of course we accept the idea to make a qualitative research even if the subject is clearly adpated to a quantitative one. saying that the objective of the paper is not to validate results but to validate if the theorical framework. is also adpated to indonesian context. to appreciate the validity of that we need many additionnal elements. the author have to make theorical proposal for this context and after develop a qualitative method to validate these proposal for indonesia.

concerning the interview guide we need an explicit table saying what are the questions and how they are related to the theorical framework

we need precise informations on the number of interviewes by category of people and their exact position and where they come from. it seems that there isn't any customer in the sample of interviewes. in this case it must be discuss about the possible biais due to that. How to be sure that different people interviewed have an objective understanding of muslim costumer behavior.

Line 250 and 251 the author speak about the primary data combined with secondary data : what are these sources of information? how they are analysed? do the author use specific coding for each one... the validity of the conclusion is impossible to confirm only from some verbatim whithout a clear understanding about how the datas from primary and secondary sources are combined to confirm the theorical framework. globaly this research isn't replicable and only the author understand the link between datas and conclusion.

Discussion arpund the determinant of non adopting decision from these customer are quite interesting and we are align on the explanation of non non adoption. the author illustrates the conclusions by some verbatim and consider their results with the perspective of research. this is positive

Author Response

See attachment - the final revision is in the manuscript.

Reviewer 2 Report

Dear Editors

Dear authors,

Thank you for the opportunity to read and review this paper. This paper describes a study that aims to examine the industrial sector 10 perspective on the factors that prevent consumers from becoming customers of Islamic banks, in 11 particular the factors that influence consumer decisions not to become Sharia bank customers. And the authors using descriptive qualitative methods to confirm and obtain input 13 from industry representatives regarding Islamic banks.

Before considering publishing this paper, this paper needs several improvements.

Introduction section

·        Lines 42-44 “Although Indonesia has the largest 42 Muslim population in the world and Islamic banking ranks fourth after Iran, Malaysia 43 and Saudi Arabia [4], it turns out that Islamic banking has failed to take advantage of this 44 demographic potential…” this statement is too subjective. Does the author mean that Muslims must choose Islamic banks? I don’t think so. people will choose banks according to quality not according to religion.

·        Lines 61-62 : author states that “On the other hand, several studies have shown that Islamic banks are quite 61 attractive in various countries with non-dominant Muslim populations…” please add some literature

Methodology section

·        please provide Ethical Considerations in this section

·        detail demographic data of respondent in needed.

·        Authors must to provide the structure of interviews and interview instrument in the manuscript. We must to check it for detail.

besides,

·        Authors must provide theoretical implication or practical implication.

·        Please separate conclusion and limitation section.

Author Response

Please see the attachment. Updates and revisions are noted with comments.

Reviewer 3 Report

I recommend the authors broaden the discussion in the introduction highlighting the contribution of this study. What are the main contributions of your research? What area of study/topic, methodology, problems identified and solved among others are your contribution to knowledge? Authors should provide a statement about the contribution of their research, underlying the new findings with contrast to the existing works. The significance of the research work done should be indicated. Furthermore, I would advise authors to stress the originality and the value-added of the paper. 

The most relevant literature on the topic at stake is contained in the paper. Yet, I recommend authors to include other relevant papers that can help give more support to their analysis. See: 

Cicchiello, A. F., Cotugno, M., Monferrã, S., & Perdichizzi, S. (2021). Exploring The Impact Of Ict Diffusion In The European Banking Industry: Evidence In The Pre-And Post-Covid-19. Journal of Financial Management, Markets and Institutions9(02), 2150010.

Implications at practical and theoretical levels should be further strengthened following the paper’s discussion. Because implications are a highly relevant element for papers published in top scholarly journals, I urge the authors to develop solid implications that help the reader to value the contributions of the study.  

Authors should pay more attention to the clarity of expression and readability. Some of the sentence construction needs to be clearer.

Round 2

Reviewer 2 Report

accepted in current form.